# Differentially Private Robust Low-Rank Approximation

**Raman Arora**
Johns Hopkins University
Baltimore, MD-21201
arora@cs.jhu.edu

**Vladimir Braverman**
Johns Hopkins University
Baltimore, MD-21201
vova@cs.jhu.edu

**Jalaj Upadhyay**
Johns Hopkins University
Baltimore, MD-21201
jalaj@jhu.edu

## Abstract

In this paper, we study the following robust low-rank matrix approximation problem: given a matrix $A \in \mathbb{R}^{n \times d}$, find a rank-$k$ matrix $M$, while satisfying differential privacy, such that $\|A - M\|_p \leq \alpha \cdot \mathsf{OPT}_k(A) + \tau$, where $\|B\|_p$ is the entry-wise $\ell_p$-norm of $B$ and $\mathsf{OPT}_k(A) := \min_{\mathsf{rank}(X) \leq k} \|A - X\|_p$. It is well known that low-rank approximation w.r.t. entrywise $\ell_p$-norm, for $p \in [1, 2)$, yields robustness to gross outliers in the data. We propose an algorithm that guarantees $\alpha = \widetilde{O}(k^2), \tau = \widetilde{O}(k^2(n + kd)/\varepsilon)$, runs in $\widetilde{O}((n + d)\mathsf{poly}\, k)$ time and uses $O(k(n + d)\log k)$ space. We study extensions to the streaming setting where entries of the matrix arrive in an arbitrary order and output is produced at the very end or continually. We also study the related problem of differentially private robust principal component analysis (PCA), wherein we return a rank-$k$ projection matrix $\Pi$ such that $\|A - A\Pi\|_p \leq \alpha \cdot \mathsf{OPT}_k(A) + \tau$.

## 1 Introduction

Low rank matrix approximation is a well studied problem, where given a data matrix $A$, the goal is to find a low-rank matrix $B$ that approximates $A$ in the sense that $\mu(A - B)$ is small under some function $\mu(\cdot)$. It finds application in numerous machine learning tasks, such as recommendation systems [10], clustering [9, 25], and learning distributions [2].

Often, the real-world data used in these applications is plagued with gross outliers, and it is desirable to impart robustness to low-rank approximation algorithms against such corruptions. Furthermore, these applications increasingly rely on sensitive data which raises the need for preserving privacy of the underlying data. The focus of this paper, therefore, is to compute a low-rank approximation of a given matrix under a strong privacy guarantee while being robust to outliers in data.

For robustness to outliers, we choose the measure $\mu(\cdot)$ to be the entrywise $\ell_p$-norm for $p \in [1, 2)$, defined as $\|A\|_p = (\sum_{i,j} |A_{i,j}|^p)^{1/p}$. It is well known that low-rank approximation w.r.t. entrywise $\ell_p$-norm, for $p \in [1, 2)$, yields robustness to gross outliers in the data [5, 7, 22, 23, 24, 29]. To address the need for privacy, we rely on the notion of differential privacy [11] that has become the *de facto* standard for private data analysis in recent years. Formally, we define differential privacy as follows.

**Definition 1.** A randomized algorithm $\mathcal{M}$ is said to be $(\varepsilon, \delta)$-differentially private if for all neighboring datasets, $A$ and $A'$, and all subsets $S \subseteq \mathsf{range}(\mathcal{M})$ in the range of $\mathcal{M}$, we have that $\Pr[\mathcal{M}(A) \in S] \leq e^\varepsilon \Pr[\mathcal{M}(A') \in S] + \delta$.

The notion of what makes two datasets neighboring determines the granularity of differential privacy [13]. At the finest scale, we consider two matrices as neighboring if they differ in at most one entry by a unit value [17, 19, 20]; this corresponds to *feature-level privacy*. At the coarsest granularity, two matrices are deemed neighboring if they differ in one row by a unit norm [18, 14]; this

corresponds to the *user-level privacy*. Note that since we do not make any boundedness assumption on the entries of the data-matrix, we need to establish a normalized scale to limit the influence of a single entry or a single row of a given matrix. In this paper, we say that two matrices $A$ and $A'$ are neighboring if the matrices are within a unit (entrywise) $\ell_1$ ball of each other, i.e., $\|A - A'\|_1 \leq 1$. This notion of neighboring datasets provides stronger guarantees than the feature-level privacy.

We are interested in private robust data analysis, specifically, robust low-rank approximation of a matrix with respect to entrywise $\ell_p$-norm for $p \in [1, 2)$, under the constraints of differential privacy. Even without privacy, low-rank matrix approximation with respect to entrywise $\ell_p$-norm for $p \neq 2$ is a non-trivial problem: it does not have a closed form solution and computing the optimal low-rank approximation with respect to $\ell_1$-norm is known to be NP-hard [16]. A natural question then is whether we can compute a good enough approximation to the best rank-$k$ approximation. This question has formed the basis for many recent results [5, 7, 22, 23, 24, 29]. However, prior to this work, differentially private low-rank approximation with respect to entrywise $\ell_p$-norm has been an open problem. We give the first time- and space-efficient differentially private algorithm for low-rank matrix approximation with respect to entrywise $\ell_p$-norm.

## 1.1 Formal Problem Statement and Contributions

In this section, we formally define the problem of differentially private robust low-rank matrix approximation, and state our main results. For the ease of presentation, we assume that $\delta = \Theta(n^{-\log n})$. We use the notation $\widetilde{O}(\cdot)$ to hide poly log factors.

**Definition 2** (Robust low-rank approximation). Given a matrix $A \in \mathbb{R}^{n \times d}$, and $p \in [1, 2)$, output a rank-$k$ matrix $M$ such that with probability at least $1 - \beta$,

$$\|A - M\|_p \leq \alpha \mathsf{OPT}_k(A) + \tau, \text{ where } \mathsf{OPT}_k(A) := \min_{\mathsf{rank}(X) \leq k} \|A - X\|_p. \tag{1}$$

Our first contribution is Algorithm 1, ROBUST-LRA, which given an input matrix $A \in \mathbb{R}^{n \times d}$ returns a differentially private rank-$k$ approximation to $A$ with a multiplicative approximation factor of $\alpha = O((k \log k)^{2(2-p)/p} \log d \log n)$ and an additive approximation error of $\tau = \widetilde{O}\left(\varepsilon^{-1} k^2 (n + kd)\right)$. In particular, for $p = 1$, we have $\alpha = O(k^2 \log^2 k \log d \log n)$ and $\tau = \widetilde{O}\left(\varepsilon^{-1} k^2 (n + kd)\right)$. We note that the best known algorithm in a non-private setting [29] achieves the same multiplicative factor, albeit with no additive error. Therefore, the price we pay for privacy is in terms of an additional additive error.

In many machine learning problems, e.g. feature selection and representation learning, all we are interested in is recovering the low-dimensional subspace spanned by the data. One such example is principal component analysis using data with gross outliers or corruptions (e.g. face recognition in the presence of occlusions). Of course, the proposed Algorithm 1 can also output the projection matrix associated with the right singular vectors of matrix $M$ with the same accuracy guarantee as for robust low-rank approximation (see Remark 1 for more details). However, the additive error we incur still scales with $n$ whereas intuitively making the basis for a $k$-dimensional subspace in $\mathbb{R}^d$ should require only adding noise proportional to $k \ll d$. This motivates a slightly different treatment for the robust principal component analysis problem, which can be formulated as follows.

**Definition 3** (Robust principal component analysis). Given a matrix $A \in \mathbb{R}^{n \times d}$, output a rank-$k$ orthonormal projection matrix $\Pi$ such that with probability at least $1 - \beta$,

$$\|A - A\Pi\|_p \leq \alpha \mathsf{OPT}_k(A) + \tau, \text{ where } \mathsf{OPT}_k(A) := \min_{\mathsf{rank}(X) \leq k} \|A - X\|_p. \tag{2}$$

The second main contribution of this paper is an algorithm that returns a differentially private rank-$k$ orthonormal projection matrix with $\alpha = O((kd \log k)^{(2-p)/p} \log^3 d \log n)$ $\tau = \widetilde{O}\left(k^2 d/\epsilon\right)$.

Many variants of differentially private low-rank approximation have been studied in the literature [14, 18, 19, 17, 20, 21, 31, 32] for both the Frobenius norm and spectral norm. We give the first $(\varepsilon, \delta)$-differentially private algorithm for robust PCA. Unlike PCA under Frobenius and spectral norm, computing an exact robust PCA is a computationally hard problem (NP-hard when $p = 1$).

Besides the objective function, our work differs from existing work also in terms of the privacy granularity and efficiency. A detailed comparison and review of previous works is presented in Table 1.

Table 1: Comparison of Models for Differentially Private $k$-Rank Approximation ($u$ and $v$ are unit vectors, $e_s$ is the $s$-th standard basis, $\eta$ is an arbitrary constant, $\omega_k := \sigma_k(A) - \sigma_{k+1}(A)$ is the singular value separation, $\mu$ is coherence of the matrix $A$, and $p \in [1, 2)$).

| | Assumptions | Accuracy $(\alpha, \tau)$ | Metric |
|---|---|---|---|
| **Theorem [10]** | $\|A - A'\|_1 = 1$ | $\left(\widetilde{O}(k^{2p(2-p)/2} \log k \log d), \widetilde{O}\left(\frac{k^2(n+kd)}{\varepsilon}\right)\right)$ | $\ell_p$-norm |
| Hardt-Roth [18] | $A - A' = e_s v^\top$ <br> $\mu$-coherence | $(\sqrt{2}, \widetilde{O}(\frac{\sqrt{kn}}{\varepsilon} + k\left(\frac{\mu\|A\|_F}{\varepsilon}\right)^{1/2}\left(\frac{d}{n}\right)^{1/4}))$ | Frobenius |
| Upadhyay [32] | $A - A' = uv^\top$ | $\left((1+\eta), \widetilde{O}\left(\varepsilon^{-1}(\sqrt{kn} + \sqrt{kd})\right)\right)$ | |
| Kapralov-Talwar [21] | $\|A\|_{op} - \|A'\|_{op} = 1$ <br> $\sigma$-value separation | $\left(1, \widetilde{O}\left((nk^3)\,\varepsilon^{-1}\right)\right)$ | Spectral |
| Hardt-Price [17] | $A - A' = e_s e_t^\top$ <br> $\mu$-coherence | $(1, \widetilde{O}(\frac{\sigma_1 \sqrt{k\mu \log(\log d\sigma_k/(\omega_k))}}{\varepsilon\omega_k}))$ | |
| Dwork et al. [14] | $A - A' = e_s v^\top$ | $\left(1, \widetilde{O}\left(\varepsilon^{-1} k\sqrt{n}\right)\right)$ | |
| Jiang et al. [20] | $A - A' = e_s e_t^\top$ | $\left(1, \widetilde{O}\left(n\varepsilon^{-1}\right)\right)$ | |

## 2 Basic Preliminaries

One of the key features of differential privacy is that it is preserved under arbitrary post-processing, i.e., an analyst, without additional information about the private database, cannot compute a function that makes an output less differentially private. This is formalized in the form of following lemma:

**Lemma 4** (Dwork *et al.* [11]). *Let $\mathcal{M}(D)$ be an $(\varepsilon, \delta)$-differential private algorithm for a data matrix $D$, and let $h$ be any function, then any mechanism $\mathcal{M}' := h(\mathcal{M}(D))$ is also $(\varepsilon, \delta)$-differentially private.*

A key ingredient in our algorithms is a $p$-stable distribution which can be defined in terms of a limit of normalized sums of i.i.d. random variables [33].

**Definition 5** ($p$-stable distirbution). *A distribution $\mathcal{D}_p$ over $\mathbb{R}$ is called $p$-stable, if there exists $p \geq 0$, such that for any $(v_1, \cdots, v_n) \in \mathbb{R}^n$, and $n$ i.i.d. random variables $X_1, \cdots, X_n$ with distribution $\mathcal{D}_p$, the random variable $\sum_i v_i X_i$ has the same distribution as the variable $\|v\|_p X$, where $X \sim \mathcal{D}_p$.*

We use the notation $\mathcal{D}_p^{(r,c)}$ to denote a distribution over $r \times c$ random matrices, where every entry of the matrix is sampled from the distribution $\mathcal{D}_p$. It is known that $p$-stable distributions exist for all $p \in (0, 2]$ [33], and that Gaussian distribution is 2-stable and the Cauchy distribution is 1-stable. Moreover, one can use the method of Chambers et al. [8] to sample from $\mathcal{D}_p$ ($1 < p < 2$).

Our analysis uses the fact that $S \sim \mathcal{D}_p^{(r,c)}$ satisfies the no-dilation and no-contraction property [28].

**Definition 6** (No-dilation [28]). *Given a matrix $A \in \mathbb{R}^{n \times d}$, if a matrix $S \in \mathbb{R}^{m \times n}$ satisfies $\|SA\|_p \leq c_1 \|A\|_p$, then $S$ has at most $c_1$ dilation on $A$ with respect to entrywise $\ell_p$-norm.*

**Definition 7** (No-contraction [28]). *Given a matrix $A \in \mathbb{R}^{n \times d}$, a matrix $S \in \mathbb{R}^{m \times n}$ has $c_2$-contraction on $A$ with respect to the entrywise $\ell_p$-norm if $\forall x \in \mathbb{R}^d, \|SAx\|_p \geq c_2^{-1} \|Ax\|_p$.*

Our analysis uses recent results from matrix sketching. In particular, we use the fact that we can approximately solve $\ell_p$-regression problem using random matrix sketches [29].

**Lemma 8** (Song et al. [29]). *Let $\Phi \in \mathbb{R}^{\phi \times n}$ be a projection matrix that preserves $\ell_p$-norm of a vector for $p \in [1, 2)$ and let $B \in \mathbb{R}^{n \times d}, C \in \mathbb{R}^{n \times c}$ be any matrix. Let $\widetilde{X} := \operatorname{argmin}_{X \in \mathbb{R}^{d \times c}} \|\Phi(BX - C)\|_p, \widehat{X} := \operatorname{argmin}_{X \in \mathbb{R}^{d \times c}} \|BX - C\|_p$, then $\|B\widetilde{X} - C\|_p \leq C_\phi \|B\widehat{X} - C\|_p$ for some constant $C_\phi$ that depends only on $\log d$.*

**Lemma 9** (Song et al. [29]). *Given matrices $L, N, A$ of appropriate dimension, let $X^* := \operatorname{argmin}_X \|LXN - A\|_p$. Suppose $S$ and $T$ satisfies $c_1$-dilation on $LX^*N - A$ and $c_2$-contraction property on $L$. Further if $\widehat{X}$ be such that $\|S(L\widehat{X}N - A)T\|_p \leq c \cdot \min_{\operatorname{rank}(X) \leq k} \|S(LXN - A)T\|_p$, then, we have that $\|L\widehat{X}N - A\|_p \leq O(c_1 c_2 c) \cdot \min_{\operatorname{rank}(X) \leq k} \|LXN - A\|_p$.*

**Algorithm 1** ROBUST-LRA

---

**Input:** Input data matrix $A \in \mathbb{R}^{n \times d}$, target rank $k$
**Output:** Rank-$k$ matrix $M \in \mathbb{R}^{n \times d}$
1: **Initialization:** Set the variables $\phi, \psi, s, t, C_\phi, C_\psi, C_s, C_t$ as in Table 2.
2: **Initialization:** Sample $\Phi \in \mathbb{R}^{\phi \times n}$, $\Psi \in \mathbb{R}^{d \times \psi}$, $S \in \mathbb{R}^{s \times n}$, and $T \in \mathbb{R}^{d \times t}$ from distributions $\mathcal{D}_p^{(\phi,n)}, \mathcal{D}_p^{(d,\psi)}, \mathcal{D}_p^{(s,n)}$, and $\mathcal{D}_p^{(d,t)}$, respectively. All these matrices are made public.
3: **Sample:** $N_1 \in \mathbb{R}^{\phi \times d}$, $N_2 \in \mathbb{R}^{n \times \psi}$, $N_3 \in \mathbb{R}^{s \times t}$ such that $N_1 \sim \mathsf{Lap}(0, C_\psi/\varepsilon)^{n \times \psi}$, $N_2 \sim \mathsf{Lap}(0, C_\phi/\varepsilon)^{\phi \times d}$, and $N_3 \sim \mathsf{Lap}(0, C_s C_t/\varepsilon)^{s \times t}$. Keep $N_1, N_2, N_3$ private.
4: **Sketch:** Compute $Y_r = \Phi A + N_1$, $Y_c = A\Psi + N_2$.
5: **Sketch:** Compute $Z_r = Y_r T$, $Z_c = SY_c$, $Z = SAT + N_3$.
6: **SVD:** Compute $[U_c, \Sigma_c, V_c] = \mathsf{SVD}(Z_c)$, $[U_r, \Sigma_r, V_r] = \mathsf{SVD}(Z_r)$.
7: $\ell_2$-**LRA:** Compute $\widehat{X} = V_c \Sigma_c^\dagger [U_c^T Z V_r^T]_k \Sigma_r^\dagger U_r^T$, where $[B]_k = \mathrm{argmin}_{r(X) \leq k} \|B - X\|_F$.
8: **Output:** $M = Y_c \widehat{X} Y_r$.

---

Table 2: Values of different variables.

| $C_\phi, C_s$ | $C_\psi, C_t$ | $\phi, \psi, s, t$ |
|---|---|---|
| $O(\log d)$ | $O(\log n)$ | $O(k \log k \log(1/\delta))$ |

## 3 Differentially private robust LRA

In this section, we give an $(\varepsilon, \delta)$-differentially private polynomial-time algorithm for robust low-rank approximation. We first discuss algorithmic challenges in extending known techniques and analyses to our problem. We present the proposed algorithm and main results in Section 3.1, and discuss extensions to the general turnstile model and the continual release model in Section 3.2. Proofs of all results are deferred to the supplementary material of this paper.

Two common approaches to preserve privacy are output perturbation [11] and input perturbation [3, 30] of the objective function. In output perturbation, we first compute the output (e.g. rank-$k$ approximation of a given matrix) non-privately and then add appropriately scaled noise to preserve privacy. In input perturbation, we add noise to the private matrix and then compute the output on the noisy matrix. Both these approaches require adding noise to every entry of the given input matrix or to every entry of the non-private output matrix. Consequently, both of these methods would incur an additive error of $O(nd)$. On the other hand, most existing non-private algorithms for robust low-rank approximation either use heuristics and do not have provable guarantees, or they make additional assumptions on the input matrix; the only exception is the work of Song et al. [29]. Again, a naive mechanism to make the algorithm of Song et al. [29] private would incur an additive error of $O(nd)$.

### 3.1 Proposed Algorithm

It is somewhat tantalizing, from a computational perspective, to attempt approximating a solution to the robust LRA problem using a low-rank approximation with respect to $\ell_2$-norm; however, it is well understood that the latter is quite sensitive to even a single outlier. A key idea behind the proposed solution then is based on the following key observation. We can approximate the output of robust low rank approximation using low rank approximation with respect to $\ell_2$-norm after sketching the matrix using $S \sim \mathcal{D}_p^{(r,n)}$ and $T \sim \mathcal{D}_p^{(c,d)}$ for some choice of $r$ and $s$. In particular, $p$-stable distribution imparts robustness, and the effect of outliers is reduced in the lower dimensional space.

In summary, the proposed algorithms are based on the following three algorithmic primitives: (a) sketching the row-space and column-space of the input matrix, (b) formulating the low-rank matrix approximation problem as a regression problem, and (c) approximating the solution to $\ell_p$ regression problem by corresponding $\ell_2$ regression problem. The analysis, then, carefully bounds the error in approximation for each of the steps above as well as error resulting from the privacy mechanism.

The pseudo-code of the proposed algorithm (ROBUST-LRA) is presented as Algorithm 1. We present values of various variables used in the algorithm in Table 2. Our main result is as follows.

**Theorem 10.** *Algorithm* ROBUST-LRA *(see Algorithm 1) is $(\varepsilon, \delta)$-differentially private. Furthermore, given a matrix $A \in \mathbb{R}^{n \times d}$, it runs in* $\mathsf{poly}(k, n, d)$ *time,* $\widetilde{O}(k(n+d))$ *space, and outputs a rank $k$ matrix $M$ such that, with probability $9/10$ over the randomness of the algorithm,*

$$\|A - M\|_p \leq O((k \log k \log(1/\delta))^{2(2-p)/p} \log d \log n) \mathsf{OPT}_k(A) + \widetilde{O}(k^2(n+kd) \log^2(1/\delta)/\varepsilon),$$

*where* $\mathsf{OPT}_k(A) := \min_{\mathsf{rank}(X) \leq k} \|A - X\|_p$.

In particular, for $p = 1$, we get

$$\|A - M\|_p \leq O(k^2 \log^2 k \log^2(1/\delta) \log d \log n) \mathsf{OPT}_k(A) + \widetilde{O}(k^2(n+kd) \log^2(1/\delta)/\varepsilon).$$

**Remark 1.** Algorithm ROBUST-LRA (Figure 1) outputs a low-rank matrix. However, it is possible to output a low-rank factorization without any loss in efficiency. It can be done by computing the SVD $[U_{\widehat{X}}, \Sigma_{\widehat{X}}, V_{\widehat{X}}]$ of $\widehat{X}$, the QR decomposition of $Y_c$ and $Y_r$ to get orthonormal bases $U$ of column space of $Y_c$ and $V$ of the row space of $Y_r$. The algorithm then outputs $[UU_{\widehat{X}}, \Sigma_{\widehat{X}}, VV_{\widehat{X}}]$ as a low-rank factorization. The extra running time of this algorithm is $O(\phi^2 d + \psi^2 n + \phi\psi^2) = \widetilde{O}(k^2(n+d))$. This is smaller than $O(nd^2)$ time if one naively factorizes $M$.

**Remark 2** (Additive Error). The additive error in Theorem 10 has a quadratic dependence on $k$. There is an implicit tradeoff between the additive and multiplicative error as $k$ increases. When $k$ is small, then error due to $\mathsf{OPT}_k(A)$ is higher, and when $k$ is larger, then the additive error is high. For instance when $k$ equals to the rank of the matrix, then we have zero multiplicative error, but additive error is of order $O(k^2 n)$. Note that $O(kn)$ error is unavoidable because we are trying to hide every single entry of the matrix $A$. Without making additional strong assumptions such as (a) stochastic data, and/or (b) incoherence, and/or (c) bounded norms, $O(kn)$ additive error is perhaps the best we can hope for. Intuitively, we have to privatize a $k$-dimensional latent representation of our data and therefore at least add noise proportional to $kn$.

## 3.2 Extension to Other Models of Differential Privacy

ROBUST LRA can be easily extended to the streaming model of computation [32] and the continual release model [12]. We first define the basic streaming model of computation that we study.

**Definition 11** (General turnstile update model). In the *general turnstile update model*, a matrix $A \in \mathbb{R}^{n \times d}$ is streamed in the form of tuple $(\Delta_t, i_t, j_t, )$, where $1 \leq i_t \leq n, 1 \leq j_t \leq d$ and $\Delta_t \in \mathbb{R}$. An update is of the form $A_{i_t, j_t} \leftarrow A_{i_{t-1}, j_{t-1}} + \Delta_t$. The curator is required to output a robust PCA or robust subspace for the matrix at the end of the stream.

For example, in the figure, the server receives an update of 6 to $A_{1,1}$ and it updates the small sketch using an update function, $U$.

At the end of the stream, the server uses the small sketch and runs an algorithm $S$ to compute the function (low-rank approximation in our context).

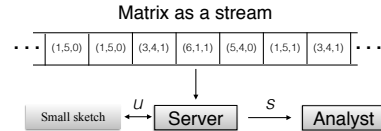

We call two streams neighboring if they are formed by neighboring matrices. Note that the private matrix is stored only in the form of linear sketches, therefore, to get an algorithm in the general turnstile streaming model, we initialize $Y_r = N_1, Y_c = N_2$, and $Z = N_3$. Then when we receive $(\Delta_t, i_t, j_t) \in \mathbb{R} \times [n] \times [d]$, we construct a matrix $A^{(t)}$ with all entries zero except for $A_{i_t, j_t}^{(t)} = \Delta_t$. We then update the sketches as follows: $Y_c = Y_c + \Phi A^{(t)}, Y_r = Y_c + A^{(t)} \Psi$, and $Z = Z + SA^{(t)}T$. Once all the updates are made, we simply run the last three steps of ROBUST-LRA. As a result, we get the following corollary.

**Corollary 12** (Informal). *Algorithm* ROBUST-LRA *is an $(\varepsilon, \delta)$-differentially private that on input a private matrix $A$ in a general turnstile update model, outputs a rank $k$ matrix $M$ with the same accuracy guarantee as in Theorem 10.*

ROBUST-LRA can also be extended to the following continual release setting [12].

**Definition 13** (Continual release model). In a *continual release model*, a matrix $A \in \mathbb{R}^{n \times d}$ is streamed in the form of tuple $(\Delta_t, i_t, j_t)$, where $1 \leq i_t \leq n, 1 \leq j \leq d$ and $\Delta_t \in \mathbb{R}$. An update is of the form $A_{i_t, j_t} \leftarrow A_{i_{t-1}, j_{t-1}} + \Delta_t$. The curator is required to output a robust PCA or robust subspace for the matrix streamed up until any time $t \leq T$.

---

**Algorithm 2** ROBUST-PCA

---

**Input:** Input data matrix $A \in \mathbb{R}^{d \times n}$, target rank $k$
**Output:** Rank-$k$ projection matrix $\Pi \in \mathbb{R}^{d \times d}$
1: **Initialization:** Set the variables $\phi, \psi, t, C_\phi, C_\psi, C_t$ as in Table 2.
2: **Initialization:** Sample $\Phi \in \mathbb{R}^{\phi \times d}, \Psi \in \mathbb{R}^{n \times \psi}, S \in \mathbb{R}^{s \times d}$, and $T \in \mathbb{R}^{n \times t}$ from distributions $\mathcal{D}_p^{(\phi,d)}, \mathcal{D}_p^{(n,\psi)}, \mathcal{D}_p^{(s,d)}$, and $\mathcal{D}_p^{(n,t)}$, respectively. All these matrices are made public.
3: **Sample:** $N_1 \in \mathbb{R}^{\phi \times t}, N_2 \in \mathbb{R}^{d \times \psi}$ such that $N_1 \sim \mathsf{Lap}(0, C_\phi C_t/\varepsilon)^{\phi \times t}, N_2 \sim \mathsf{Lap}(0, C_\psi/\varepsilon)^{d \times \psi}$. Keep $N_1, N_2$ private.
4: **Sketch:** Compute $Y_r = \Phi A^T T + N_1$ and $Y_c = A^T \Psi + N_2$. $Z_c = \Phi Y_c$ and $Z = Y_r$.
5: **SVD:** Compute $[U_c, \Sigma_c, V_c] = \mathsf{SVD}(Z_c)$,
6: $\qquad\qquad [U_r, \Sigma_r, V_r] = \mathsf{SVD}(Y_r)$.
7: $\ell_2$-**LRA:** Compute $\widehat{X} = V_c \Sigma_c^\dagger [U_c^T Z V_r^T]_k \Sigma_r^\dagger U_r^T$, where $[B]_k = \operatorname{argmin}_{r(X) \leq k} \|B - X\|_F$.
8: **Pick:** a permutation matrix $Q \in R^{\phi \times \phi}$.
9: **Compute:** the full SVD of $Y_c \widehat{X}$, $[U', \Sigma', V']$. Set $U = U'Q$, $\Sigma = \Sigma'Q$, and $P = \Phi^\dagger (U\Sigma)^\dagger$.
10: **Output:** $\Pi = PU\Sigma(\Phi PU\Sigma)^\dagger \Phi$.

---

For outputting a low-rank approximation in the continual release model, we can use the generic transformation to store a binary tree that is constructed over the privatized sketches of the updates as its leaves [12]. When a new query for a range of updates is made, we accumulate the sketches of the dyadic partition of the range to compute the sketches for that range. We then compute the last three steps of ROBUST-LRA. We have the following result.

**Corollary 14.** *Algorithm* ROBUST-LRA *is an $(\varepsilon, \delta)$-differentially private algorithm that on input matrix $A$ in a streaming manner, runs in time $\mathsf{poly}(k, n, d, \log T)$ and outputs a rank $k$ matrix $M^{(t)}$ in the continual release model over $T$ time epochs, such that, with probability at least $9/10$,*
$$\|A^{(t)} - M^{(t)}\|_p \leq O((k \log k \log(1/\delta))^{2(2-p)/p} \log d \log n) \mathsf{OPT}_k(A^{(t)}) + \widetilde{O}(k^2(n + kd) \log T),$$
*where $\mathsf{OPT}_k(A)$ is as in Theorem 10, and $A^{(t)}$ is the matrix up to $t$ time epochs.*

# 4 Differentially Private Robust Principal Component Analysis

In this section, we focus on the problem of robust PCA under the constraints of differential privacy. We first present the proposed algorithm and then discuss extensions to the general turnstile model continual release model. Proofs of all results are deferred to the supplementary material of this paper.

The key ideas underlying the proposed algorithm, ROBUST-PCA (see Algorithm 2 for the pseudocode), and its analysis, essentially follow the techniques developed in the previous section for ROBUST-LRA, but with a couple of small modifications to get a better additive error. First, we only generate two sketches, $Y_r = \Phi A^T T + N_1$ and $Y_c = A^T \Psi + N_2$, where $\Psi, \Phi, T$ are random sketching matrices and $N_1, N_2$ are noise matrices as defined in Algorithm 2. Second, we solve a slightly different optimization problem:
$$\min_{\mathsf{rank}(Y) \leq k} \left\| A^T - (PU\Sigma)Y(\Phi A^T) \right\|_F,$$

where $P, U, \Sigma$ are as formed in Algorithm 2. We show that $(\Phi U\Sigma P)^\dagger$ is an approximate solution to $\min_X \left\| \Phi(A^T - PU\Sigma X \Phi A^T) T \right\|_p$. The rest of the proof then follows the same steps as in the proof of Theorem 10. In addition, we also show that $\Pi$ is an orthonormal rank-$k$ projection matrix. The above exposition focuses on the non-private setting for the sake of simplicity. The proof is more involved due to noise matrices added for privacy.

We show the following guarantee for the proposed algorithm.

**Theorem 15.** *Algorithm* ROBUST-PCA, *(see Algorithm 2), is $(\varepsilon, \delta)$-differentially private. Further, given a matrix $A \in \mathbb{R}^{n \times d}$ with $\mathsf{OPT}_k(A) := \min_{\mathsf{rank}(X) \leq k} \|A - X\|_p$, it runs in time $\mathsf{poly}(k, n, d)$, space $\widetilde{O}(k(n + d))$, and outputs a rank $k$ orthonormal projection matrix $\Pi$ such that, with probability $9/10$ over the random coin tosses of the algorithm,*

$$\|A - A\Pi\|_p \leq O((k \log k \log(1/\delta))^{2(2-p)/p} \log n \log^3 d) \mathsf{OPT}_k(A) + \widetilde{O}(k^2 d \log n/\varepsilon).$$

In particular, when $p = 1$, we have the following guarantee:

$$\|A - A\Pi\|_p \leq O(k^2 \log n \log^3 d \log^2 k \log^2(1/\delta))\mathsf{OPT}_k(A) + \widetilde{O}(k^2 d \log n/\varepsilon).$$

We note that ROBUST-PCA yields a smaller additive error than ROBUST-LRA by a factor of $n/d$, but at the expense of an additional multiplicative factor of $\log^2(d)$. Therefore, in settings where $\mathsf{OPT}_k(A)$ is small (e.g. when $A$ is nearly low rank), ROBUST-PCA enjoys a much better accuracy guarantee.

**Extension to Other Models of Differential Privacy.** We can extend ROBUST-PCA to the streaming model of computation [32] and the continual release model [12] as in Section 3.2. We can also extend ROBUST-PCA to the local model of differential privacy. Local differential privacy has gained a lot of attention recently [1, 15]. In the local privacy model, there is no central database of private data. Instead, each individual has its own data element (a database of size one), and sends a report based on its own datum in a differentially private manner.

Formally, we consider the database $X = [x_1, \cdots, x_n]^\top$ as a collection of $n$ elements (rows) from some domain $\mathcal{X} \subseteq \mathbb{R}^d$, with each row held by a different individual. The $i^{th}$ individual has access to $\varepsilon_i$-*local randomizer*, $R_i : \mathcal{X} \to W$, which is an $\varepsilon_i$-differentially private algorithm that takes as input a database of size $n = 1$. We assume that the algorithms may interact with the database only through local randomizers. We can then define local differential privacy as follows [13]. An algorithm is $\varepsilon$-locally differentially private if it accesses the database $X$ via the local randomizers, $R_1(x_1), \ldots, R_n(x_n)$, where $\max \{\varepsilon_1, \ldots, \varepsilon_n\} \leq \varepsilon$.

We note that what we have defined above is a non-interactive local differential privacy algorithm where an individual only sends a single message to the server. It was argued in Smith et al. [27] that it is more desirable to have as few rounds of interactions as possible from an implementation point of view. In fact, existing large-scale deployments are limited to one that are non-interactive. Therefore, we limit our study to what is possible in the non-interactive variant of local differential privacy.

We extend Algorithm 2 to an $\varepsilon$-locally-differentially private protocol, LOCAL-ROBUST-PCA, where every user $1 \leq i \leq n$ has a row $A_{i:}$ of the data matrix and sends only one message to the server. We show that the output produced by the server after a run of LOCAL-ROBUST-PCA is a rank-$k$ orthonormal projection matrix $\Pi \in \mathbb{R}^{d \times d}$ such that

$$\|A - A\Pi\|_p \leq O(\log n \log^3 d \, (k \log k \log(1/\delta))^{2(2-p)/p})\mathsf{OPT}_k(A) + \widetilde{O}(k^2 nd/\varepsilon).$$

The above guarantee is non-trivial when $\|A\|_p \gg nd$. Such an assumption is often valid in practical settings with large corruption to data matrices.

# 5 Discussion

In this paper, we present differentially private algorithms for robust low-rank approximation and for robust principal component analysis. In addition, we study extensions of our algorithms to a continual release model, the streaming model of computation, and the local model of differential privacy.

The bounds we provide involve a multiplicative factor that depends on the target rank $k$. Such a dependence was deemed necessary in non-private settings. In particular, Song et al. [29] show that if the exponential time hypothesis is true, then any linear-sketch based polynomial time algorithm for robust rank-$k$ factorization incurs an $\Omega(k^{1/2-\gamma})$ multiplicative approximation for some $\gamma \in (0, 0.5)$ that can be arbitrarily small. It is not clear immediately if such a result still holds when we allow an additive error in the approximation, as is the case here.

# Acknowledgements

This research was supported in part by NSF BIGDATA grant IIS-1546482, NSF BIGDATA grant IIS-1838139, NSF Career CCF-1652257, and ONR Award N00014-18-1-2364.

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
