[Supplementary Material · Supplementary Material.pdf]

# Supplementary Material to "Differentially Private Robust Low-Rank Approximation"

## A   Auxiliary Lemma

We need the following results about product of pseudo-inverse.

**Fact 16.** *If $A$ has a left-inverse, then $A^\dagger = (A^\mathsf{T} A)^{-1} A^\mathsf{T}$ and if $A$ has right-inverse, then $A^\dagger = A^\mathsf{T} (AA^\mathsf{T})^{-1}$.*

**Theorem 17** (Product of pseudoinverse). *Let $A$ and $B$ be conforming matrices and either,*

1. *$A$ has orthonormal columns (i.e., $A^\mathsf{T} A$ is an identity matrix) or,*

2. *$B$ has orthonormal rows (i.e., $BB^\mathsf{T}$ is an identity matrix),*

3. *$A$ has all columns linearly independent (full column rank) and $B$ has all rows linearly independent (full row rank) or,*

4. *$B = A^\mathsf{T}$ (i.e., $B$ is the conjugate transpose of $A$),*

*then $(AB)^\dagger = B^\dagger A^\dagger$.*

The following lemma follows from Holder's inequality and minimality of $\hat{x}$.

**Lemma 18** ($\ell_2$ relaxation of $\ell_p$ regression). *Let $p \in [1,2)$. For any $A \in \mathbb{R}^{n \times d}$ and $b \in \mathbb{R}^n$, define $x^* = \operatorname{argmin} \|Ax - b\|_p$ and $\hat{x} = \operatorname{argmin} \|Ax - b\|_2$. Then, $\|Ax^* - b\|_p \leq \|A\hat{x} - b\|_p \leq n^{1/p-1/2} \|Ax^* - b\|_p$.*

**Lemma 19** (Subsampling and rescaling lemma). *Let $k$ be a parameter and $s = O(k \log k)$. Let $S \in \mathbb{R}^{s \times d}$ be a random matrix with every entries sampled i.i.d. from $C(0,1)$, Cauchy distribution with variance 1, and scaled by $1/s$. Let $A, B \in \mathbb{R}^{d \times p}$ be real-valued matrices such that $\operatorname{rank}(A) \leq k$. Let $S' \in \mathbb{R}^{s \times d}$ be a matrix with $i$-th row $S'_{i:}$ defined using the following probability distribution*

$$S'_{i:} = \begin{cases} \frac{s}{k} S_{i:} & \text{with probability } k/s \\ 0^d & \text{otherwise} \end{cases}$$

*Then with probability $24/25$, $\|S'A - SB\|_1 \leq O(\log d) \|A - B\|_1$.*

*Proof.* Let $\tilde{p} = sp$. We use the notation $C(0,1)$ to define Cauchy random variable with variance 1. We call a random variable $c$ *half clipped Cauchy random variable* if $c \sim |C(0,1)|$. Define $\mathsf{G}_u$ to be event when half clipped random Cauchy variable $c_u$, such that $c_u < 100\tilde{p}$. Let $\mathsf{Good} = \cap \mathsf{G}_u$. We can easily compute the probability that $\mathsf{G}_u$ and Good happens. Using the pdf of Cauchy, we have

$$\Pr[\mathsf{G}_u] = 1 - \frac{2}{\pi} \tan^{-1}(1/100sp) \geq 1 - \frac{1}{50\pi\tilde{p}}.$$

Union bound then implies that $\Pr[\neg \mathsf{Good}] \leq \frac{1}{50\pi}$. Using total probability theorem and Markov's inequality, we have

$$\Pr[\|S'A - SB\|_1 \geq (100 \log p) \|A - B\|_1] \leq \Pr[\|S'A - SB\|_1 \geq (100 \log p) \|A - B\|_1 \,|\, \mathsf{Good}]$$
$$+ \Pr[\neg \mathsf{Good}] \leq \frac{\mathsf{E}[\|S'A - SB\|_1 \,|\, \mathsf{Good}]}{100 \log p \|A - B\|_1} + \frac{1}{50\pi} \tag{3}$$

Let $A_{:i}$ denote the $i$-th column of the matrix $A$. Then

$$\mathsf{E}[\|S'A - SB\|_1 \,|\mathsf{Good}] = \sum_{i=1}^{p} \mathsf{E}[\|S'A_{:i} - SB_{:i}\|_1 \,|\mathsf{Good}]$$

$$= \sum_{i=1}^{p} \mathsf{E}\left[\sum_{j=1}^{s}\left|\sum_{\ell=1}^{d} \frac{1}{s} S'_{j,\ell} A_{\ell,i} - S_{j,\ell} B_{\ell,i}\right| \,|\mathsf{Good}\right]$$

$$= \frac{1}{s} \sum_{i=1}^{p} \sum_{j=1}^{s} \mathsf{E}[\bar{s}_{i,j}|\mathsf{Good}]$$

$$= \frac{1}{s} \sum_{i=1}^{p} \sum_{j=1}^{s} \|A_{:i} - B_{:i}\| \,\mathsf{E}[c_{i+p(j-1)}|\mathsf{Good}], \tag{4}$$

where $\bar{s}_{i,j} \sim |C(0, \|A_{:i} - B_{:i}\|_1)|$ and $c_{i+p(j-1)}$ is half clipped Cauchy random variable, i.e., $c_{i+p(j-1)} \sim |C(0,1)|$. We next compute $\mathsf{E}[c_{i+p(j-1)}|\mathsf{Good}]$. Let $u = i + p(j-1)$.

Since for any random variable $X$ and $Y$, $\mathsf{E}[X] = \sum_y \Pr[Y=y]\mathsf{E}[X|Y=y])$, we have

$$\mathsf{E}[c_u|\mathsf{G}_u] = \Pr[\mathsf{Good}|\mathsf{G}_u]\mathsf{E}[c_u|\mathsf{G}_u \cap \mathsf{Good}] + \Pr[\neg\mathsf{Good}|\mathsf{G}_u]\mathsf{E}[c_u|\mathsf{G}_u \cap \neg\mathsf{Good}]$$

$$\geq \mathsf{E}[c_u|\mathsf{G}_u \cap \mathsf{Good}]\Pr[\mathsf{Good}|\mathsf{G}_u] = \mathsf{E}[c_u|\mathsf{Good}]\Pr[\mathsf{Good}|\mathsf{G}_u],$$

In other words,

$$\mathsf{E}[c_u|\mathsf{Good}] \leq \frac{\mathsf{E}[c_u|\mathsf{G}_u]\Pr[\mathsf{G}_u]}{\Pr[\mathsf{Good}|\mathsf{G}_u]}.$$

Now $\mathsf{E}[c_u|\mathsf{G}_u] = \frac{\log(1+(sp)^2)}{\pi\Pr[\mathsf{G}_u]}$. Using Bayes theorem and the fact that $\Pr[\mathsf{Good}] = \Pr[\mathsf{Good} \cap \mathsf{G}_u]$, this implies that

$$\mathsf{E}[c_u|\mathsf{Good}] \leq \frac{\mathsf{E}[c_u|\mathsf{G}_u]\Pr[\mathsf{G}_u]}{\Pr[\mathsf{Good}]} = \frac{\log(1+(sp)^2)}{\pi\Pr[\mathsf{G}_u]}\frac{\Pr[\mathsf{G}_u]}{\Pr[\mathsf{Good}]}$$

$$\leq \frac{\log(1+(sp)^2)}{\pi(1-1/(50\pi))} \leq 2\log(sp).$$

We can now bound equation (4) as below:

$$\mathsf{E}\left[\|S'A - SB\|_1 \,|\mathsf{Good}\right] = \frac{1}{s}\sum_{i=1}^{p}\sum_{j=1}^{s}\|A_{:i} - B_{:i}\|_1 \,\mathsf{E}[c_{i+p(j-1)}|\mathsf{Good}] = 2\log p\|A - B\|_1.$$

Plugging this in equation (3), we have

$$\Pr[\|S'A - SB\|_1 \geq 100\log p\,\|A-B\|_1] \leq \frac{\mathsf{E}[\|S'A-SB\|_1\,|\mathsf{Good}]}{50\log p\,\|A-B\|_1} + \frac{1}{5\pi}$$

$$\leq \frac{2\log sp}{100\log p} + \frac{1}{50\pi} \leq \frac{1}{25},$$

where the last inequality holds because $s \leq p$. This completes the proof. $\qquad\square$

## B  Missing Proofs

### B.1  Proof of Theorem 10

**Reminder of Theorem 10.** Algorithm Robust-LRA, (see Algorithm 1), is $(\varepsilon, \delta)$-differentially private. Furthermore, given a matrix $A \in \mathbb{R}^{n\times d}$, it runs in $\mathrm{poly}(k,n,d)$ time, $\widetilde{O}(k(n+d))$ space, and outputs a rank $k$ matrix $M$ such that, with probability $9/10$ over the randomness of the algorithm,

$$\|A - M\|_p \leq O((k\log k\log(1/\delta))^{2(2-p)/p}\log d\log n)\mathsf{OPT}_k(A) + \widetilde{O}(k^2(n+kd)\log^2(1/\delta)/\varepsilon),$$

*Proof of Theorem 10.* We first give the privacy proof of Theorem 10. Let $A$ and $A'$ be neighboring matrices, i.e., $\|A - A'\|1 \leq 1$. We argue the privacy result for $p = 1$ (the case for $p \in (1,2)$ follows from invoking Holder's inequality. The private matrix is used to generate three sketches: $Y_c, Y_r$, and $Z$. Since $\Phi, \Psi, S$, and $T$ are sampled from distribution of random matrices that preserves the $\ell_1$-norm, we have that $\|\Phi(A - A')\|1 \leq C_\phi \|A - A'\|1 = C_\phi$ with probabilty at least $1 - \delta$. The privacy proof now follows from Laplace mechanism. Note that for $p \in (1,2)$, we have $\|\Phi(A - A')\|_p \leq C_\phi \|(A - A')\|_p \leq C_\phi \|(A - A')\|_1$.

We now give the utility proof of Theorem 10. Let

$$U^*, V^* := \underset{\substack{U \in \mathbb{R}^{n \times k} \\ V \in \mathbb{R}^{k \times d}}}{\operatorname{argmin}} \|UV - A\|_p$$

Our proof relies on three fundamental techniques.

**Two fundamental techniques.** The first fundamental technique is to use the fact that solving generalized linear regression problem in the projected space gives an approximate solution to the original generalized regression problem. The second main idea is the reduction from low-rank approximation to a generalized linear regression problem.

Let $B = A + S^\dagger N_3 T^\dagger$, then $SBT = Z$. Also let $C = A + \Phi^\dagger N_1$, then $\Phi C = Y_r$. Let

$$\widetilde{V} := \underset{V \in \mathbb{R}^{k \times d}}{\operatorname{argmin}} \|\Phi(U^* V - C)\|_p,$$

$$\widehat{V} := \underset{V \in \mathbb{R}^{k \times d}}{\operatorname{argmin}} \|\Phi(U^* V - C)\|_F,$$

$$V' := \underset{V \in \mathbb{R}^{k \times d}}{\operatorname{argmin}} \|U^* V - C\|_p$$

Then using Lemma 8 and the fact that $\|U^* V' - B)\|_p \leq \|U^* V - B)\|_p$ for all $V$ (and in particular, $V^*$), we have

$$\left\| U^* \widetilde{V} - C) \right\|_p \leq O(C_\phi) \|U^* V' - C)\|_p$$

$$\leq O(C_\phi) \left( \|U^* V^* - A)\|_p + \left\| \Phi^\dagger N_1 \right\|_p \right). \tag{5}$$

Since $\widehat{V}_{:i} = (\Phi U^*)^\dagger \Phi C_{:i} = \operatorname{argmin}_x \|\Phi(U^* x - C_{:i})\|_F$, using Holder's inequality, we have

$$\left\| (U^* \widehat{V} - C) \right\|_p = \sum_{i=1}^{d} \left\| (U^* \widehat{V}'_{:i} - C_{:i}) \right\|_p$$

$$\leq \sqrt{\phi} \sum_{i=1}^{d} \left\| (U^* \widetilde{V}_{:i} - C) \right\|_p \qquad \text{(Lemma 18)}$$

$$= \sqrt{\phi} \left\| U^* \widetilde{V} - C) \right\|_p.$$

Combining this with equation (5), we have

$$\left\| (U^* \widehat{V} - C) \right\|_p \leq O(C_\phi \sqrt{\phi}) \|U^* V^* - A)\|_p + O(C_\phi \sqrt{\phi}) \left\| \Phi^\dagger N_1 \right\|_p.$$

Moreover,

$$\left\| (U^* \widehat{V} - C) \right\|_p \geq \left\| U^* \widehat{V} - A) \right\|_p - \left\| \Phi^\dagger N_1 \right\|_p.$$

Combining the last two inequalities gives us

$$\left\| (U^* \widehat{V} - A) \right\|_p \leq O(C_\phi \sqrt{\phi}) \|U^* V^* - A)\|_p + O(C_\phi \sqrt{\phi}) \left\| \Phi^\dagger N_1 \right\|_p. \tag{6}$$

Further let,

$$\widetilde{U} := \operatorname*{argmin}_{U \in \mathbb{R}^{n \times k}} \left\| (U\widehat{V} - A)\Psi \right\|_p,$$

$$\widehat{U} := \operatorname*{argmin}_{U \in \mathbb{R}^{n \times k}} \left\| (U\widehat{V} - A)\Psi \right\|_F,$$

$$U' := \operatorname*{argmin}_{U \in \mathbb{R}^{n \times k}} \left\| U\widehat{V} - A \right\|_p$$

Then using Lemma 8 and the fact that $\left\| U'\widehat{V} - B) \right\|_p \leq \left\| U\widehat{V} - B) \right\|_p$ for all $U$ (and in particular, $U^*$), we have

$$\left\| \widetilde{U}\widehat{V} - A) \right\|_p \leq O(C_\psi) \left\| U'\widehat{V} - A) \right\|_p$$
$$\leq O(C_\psi) \left\| (U^*\widehat{V} - A) \right\|_p. \tag{7}$$

We know that $\widehat{U}_{i:} = A_{:i}\Psi(\widehat{V}\Psi)^\dagger = \operatorname{argmin}_x \left\| (x\widehat{V} - A_{i:})\Psi \right\|_F$. Equation (7) then gives us

$$\left\| (\widehat{U}\widehat{V} - A) \right\|_p = \sum_{i=1}^{n} \left\| (\widehat{U}_{i:}\widehat{V} - A_{i:}) \right\|_p$$

$$\leq \sqrt{\psi} \sum_{i=1}^{d} \left\| (\widetilde{U}_{i:}\widehat{V} - A_{i:}) \right\|_p \qquad \text{(Lemma 18)}$$

$$= \sqrt{\psi} \left\| \widetilde{U}\widehat{V} - A) \right\|_p$$

$$\leq O(C_\psi \sqrt{\psi}) \left\| (U^*\widehat{V} - A) \right\|_p. \tag{8}$$

Substituting the value of $\widehat{U} = A\Psi(\widehat{V}\Psi)^\dagger$,

$$\left\| A\Psi(\widehat{V}\Psi)^\dagger\widehat{V} - A \right\|_p \leq O(C_\phi C_\psi \sqrt{\psi\phi}) \left\| U^*V^* - A \right\|_p + O(C_\phi C_\psi \sqrt{\psi\phi}) \left\| \Phi^\dagger N_1 \right\|_p \tag{9}$$

Recall that $Y_c = A\Psi + N$ by the construction in the algorithm. Using subadditivity of norms and substituting $\widehat{V} = (\Phi U^*)^\dagger Y_r$, we have

$$\left\| Y_c(\widehat{V}\Psi)^\dagger\widehat{V} - A \right\|_p \leq \left\| A\Psi(\widehat{V}\Psi)^\dagger\widehat{V} - A \right\|_p + \left\| N(\widehat{V}\Psi)^\dagger\widehat{V} \right\|_p \qquad \text{(subadditivity)}$$

$$\leq O(C_\phi C_\psi \sqrt{\psi\phi}) \left\| U^*V^* - A) \right\|_p + \left\| N_2(\widehat{V}\Psi)^\dagger\widehat{V} \right\|_p \qquad \text{(equation (9))}$$

$$+ O(C_\phi C_\psi \sqrt{\psi\phi}) \left\| \Phi^\dagger N_1 \right\|_p \tag{10}$$

Now again from subadditivity, we have

$$\left\| Y_c(\widehat{V}\Psi)^\dagger\widehat{V} - B \right\|_p \leq \left\| Y_c(\widehat{V}\Psi)^\dagger\widehat{V} - A \right\|_p + \left\| S^\dagger N_3 T^\dagger \right\|_p$$

Combining equation (10) with the above inequality, we get

$$\left\| Y_c(\widehat{V}\Psi)^\dagger\widehat{V} - B \right\|_p \leq O(C_\phi C_\psi \sqrt{\psi\phi}) \left\| U^*V^* - A) \right\|_p$$
$$+ \left\| N_2(\widehat{V}\Psi)^\dagger\widehat{V} \right\|_p + \left\| S^\dagger N_3 T^\dagger \right\|_p$$
$$+ O(C_\phi C_\psi \sqrt{\psi\phi}) \left\| \Phi^\dagger N_1 \right\|_p.$$

Further, since $U^*$ has rank at most $k$ and $\widehat{V} = (\Phi U^*)^\dagger Y_r$, $(\widehat{V}\Psi)^\dagger(\Phi U^*)^\dagger\Phi$ has rank at most $k$. This implies that

$$\min_{r(X)\leq k}\|Y_c X Y_r - B\|_p \leq \left\|Y_c(\widehat{V}\Psi)^\dagger(\Phi U^*)^\dagger Y_r - B\right\|_p \qquad \text{(minimality)}$$

$$\leq O(C_\phi C_\psi \sqrt{\psi\phi})\|U^*V^* - A)\|_p + \left\|N_2(\widehat{V}\Psi)^\dagger\widehat{V}\right\|_p$$

$$+ \left\|S^\dagger N_3 T^\dagger\right\|_p + O(C_\phi C_\psi \sqrt{\psi\phi})\left\|\Phi^\dagger N_1\right\|_p \qquad (11)$$

**Third fundamental technique.** The last fundamental technique that we use is that an approximate solution of low-rank problem in the projected space also gives an approximate solution of the original low-rank problem. Let $Q = SAT$ and

$$\widehat{X} = V_c \Sigma_c^\dagger [U_c^T Z V_r^T]_k \Sigma_r^\dagger U_r^T.$$

Let $\widetilde{X} := \operatorname{argmin}_{\mathsf{rank}(Y)\leq k}\|SY_c X Y_r T - Z\|_p$. To show that we can achieve an approximate solution of a low-rank problem in the projected space, we use Holder's inequality. More precisely, we have the following set of inequalities:

$$\left\|SY_c\widehat{X}Y_r T - Z\right\|_p \leq \sqrt{st}\left\|SY_c\widehat{X}Y_r T - Z\right\|_F$$

$$= \sqrt{st}\min_{\mathsf{rank}(Y)\leq k}\left\|SY_c\widehat{X}Y_r T - Z\right\|_F \qquad \text{(by definition)}$$

$$\leq \sqrt{st}\left\|SY_c\widetilde{X}Y_r T - Z\right\|_F \qquad \text{(by minimality)}$$

$$\leq \sqrt{st}\left\|SY_c\widetilde{X}Y_r T - Z\right\|_p$$

$$= \sqrt{st}\min_{\mathsf{rank}(Y)\leq k}\|SY_c X Y_r T - Z\|_p, \qquad (12)$$

where the first and last inequalites follow from Holder's inequality, second inequality from the minimality, the first equality is due to [32], and the last equality is by definition. Using Lemma 9, we have

$$\left\|Y_c\widehat{X}Y_r - B\right\|_p \leq \widetilde{O}(\sqrt{st})\min_{\mathsf{rank}(X)\leq k}\|Y_c X Y_r - B\|_p.$$

Now, we have from subadditivity of norm,

$$\left\|Y_c\widehat{X}Y_r - A\right\|_p - \left\|S^\dagger N_3 T^\dagger\right\|_p \leq \left\|Y_c\widehat{X}Y_r - B\right\|_p.$$

Combining this with equation (11), we have

$$\left\|Y_c\widehat{X}Y_r - B\right\|_p \leq \widetilde{O}(C_\phi C_\psi \sqrt{\psi\phi st})\|U^*V^* - A)\|_p + \sqrt{st}\left\|N_2(\widehat{V}\Psi)^\dagger\widehat{V}\right\|_p$$

$$+ 2\sqrt{st}\left\|S^\dagger N_3 T^\dagger\right\|_p + O(C_\phi C_\psi \sqrt{\psi\phi st})\left\|\Phi^\dagger N_1\right\|_p \qquad (13)$$

Note that $Y_c\widehat{X}Y_r$ is the output of the algorithm. Therefore, all that remain is to bound each of the above additive term. The following claim does this.

**Claim 20.** *With probability at least* $24/25$,

$$\left\|\Phi^\dagger N_1\right\|_p \leq \widetilde{O}(C_\phi d\phi/\varepsilon),$$

$$\left\|N_2(\widehat{V}\Psi)^\dagger\widehat{V}\right\|_p \leq \widetilde{O}(C_\psi kn/\varepsilon),$$

$$\left\|S^\dagger N_3 T^\dagger\right\|_p \leq \widetilde{O}(C_s C_t st/\varepsilon).$$

*Proof.* Now $N_2(\widehat{V}\Psi)^\dagger \widehat{V}\Psi = \widehat{N}_2$. Using no dilation property of $\Phi$, we have $\left\|N_2(\widehat{V}\Psi)^\dagger \widehat{V}\right\|_p \leq$ $O(C_\psi)\left\|\widehat{N}_2\right\|_p$. This can be bound using the standard tail inequality for Laplace mechanism, i.e, with probability at least $99/100$, $\left\|\widehat{N}_1\right\|_p = \widetilde{O}(kn)$. Similarly, $\Phi\Phi^\dagger N_1 = \widehat{N}_1$ and $SS^\dagger N_3 T^\dagger T = N_3$. Using dilation and contraction properties of $\Phi, \Psi, S$, and $T$ completes the proof of claim. $\qquad\square$

Using Claim 20 in equation (13) completes the proof of Theorem 10. $\qquad\square$

## B.2 Proof of Theorem 15

**Restatement of Theorem 15.** Algorithm ROBUST-PCA, (see Algorithm 2), is $(\varepsilon, \delta)$-differentially private. Further, given a matrix $A \in \mathbb{R}^{n \times d}$ with $\mathsf{OPT}_k(A) := \min_{\mathrm{rank}(X) \leq k} \|A - X\|_p$, it runs in time $\mathrm{poly}(k, n, d)$, space $\widetilde{O}(k(n + d))$, and outputs a rank $k$ orthonormal projection matrix $\Pi$ such that, with probability $9/10$ over the random coin tosses of the algorithm,

$$\|A - A\Pi\|_p \leq O((k \log k \log(1/\delta))^{2(2-p)/p} \log n \log^3 d)\mathsf{OPT}_k(A) + \widetilde{O}(k^2 d \log n / \varepsilon).$$

*Proof of Theorem 15.* We start by giving the privacy proof. Let $A$ and $A'$ be neighboring matrices, i.e., $\|A - A'\|_1 \leq 1$. We argue the privacy result for $p = 1$ (the case for $p \in (1, 2)$ follows from invoking Holder's inequality. The private matrix is used to generate two sketches: $Y_c, Y_r$. Since $\Phi, \Psi$, and $T$ are sampled from distribution of random matrices that preserves the $\ell_p$-norm, we have that $\|\Phi(A - A')\|_1 \leq C_\phi \|A - A'\|_1 = C_\phi$ with probabilty at least $1 - \delta$. The privacy proof now follows from Laplace mechanism. Note that for $p \in (1, 2)$, we have $\|\Phi(A - A')\|_p \leq C_\phi \|(A - A')\|_p \leq C_\phi \|(A - A')\|_1$.

We now move to prove the utility guarantee. For the ease of presentation, we just present the case for $p = 1$. The case for $p \in (1, 2)$ follows similarly.

Let us define $\widehat{X} = V_c \Sigma_c^\dagger [U_c^T Z V_r^T]_k \Sigma_r^\dagger U_r^T$. Further, let

$$U^*, V^* := \underset{\substack{U \in \mathbb{R}^{n \times k} \\ V \in \mathbb{R}^{k \times d}}}{\mathrm{argmin}} \|UV - A)\|_p$$

**Two fundamental techniques.** The first fundamental technique is to use the fact that solving generalized linear regression problem in the projected space gives an approximate solution to the original generalized regression problem. Then we use the reduction from low-rank approximation to a generalized linear regression problem.

Let $B = A^T + \Phi^\dagger N_1 T^\dagger$, then $\Phi B T = Z$. We now define the following optimization problems:

$$\widetilde{V} := \underset{V \in \mathbb{R}^{k \times d}}{\mathrm{argmin}} \|\Phi(U^*V - B)\|_p,$$

$$\widehat{V} := \underset{V \in \mathbb{R}^{k \times d}}{\mathrm{argmin}} \|\Phi(U^*V - B)\|_F,$$

$$V' := \underset{V \in \mathbb{R}^{k \times d}}{\mathrm{argmin}} \|U^*V - B)\|_p$$

Then using Lemma 8 and the fact that $\|U^*V' - B)\|_p \leq \|U^*V - B)\|_p$ for all $V$ (and in particular, $V^*$), we have

$$\left\|U^*\widetilde{V} - B\right\|_p \leq O(\log d) \|U^*V' - B)\|_p \qquad\qquad \text{(Lemma 8)}$$

$$\leq O(\log d) \left\|U^*V^* - A^T)\right\|_p + O(\log d) \left\|\Phi^\dagger N_1 T^\dagger\right\|_p.$$

Since $\widehat{V}_{:i} = (\Phi U^*)^{\dagger} \Phi B_{:i} = \min_x \|\Phi(U^* x - B_{:i})\|_F$, using Holder's inequality, we have

$$\left\|(U^*\widehat{V} - B)\right\|_p = \sum_{i=1}^{d} \left\|(U^*\widehat{V}'_{:i} - B_{:i})\right\|_p$$

$$\leq \sqrt{\phi} \sum_{i=1}^{d} \left\|(U^*\widetilde{V}_{:i} - B)\right\|_p \qquad \text{(Lemma 18)}$$

$$= \sqrt{\phi} \left\|U^*\widetilde{V} - B)\right\|_p. \qquad (14)$$

In other words,

$$\left\|(U^*\widehat{V} - B)\right\|_p \leq O(\sqrt{\phi} \log d) \left\|U^*V^* - A^T)\right\|_p + O(\sqrt{\phi} \log d) \left\|\Phi^{\dagger} N_1 T^{\dagger}\right\|_p.$$

Moreover,

$$\left\|(U^*\widehat{V} - B)\right\|_p \geq \left\|U^*\widehat{V} - A^T)\right\|_p + \left\|\Phi^{\dagger} N_1 T^{\dagger}\right\|_p.$$

Combining the last two inequalities gives us

$$\left\|(U^*\widehat{V} - A^T)\right\|_p \leq O(\sqrt{\phi} \log d) \left\|U^*V^* - A^T)\right\|_p + O(\sqrt{\phi} \log d) \left\|\Phi^{\dagger} N_1 T^{\dagger}\right\|_p. \qquad (15)$$

Now define the following optimization problems:

$$\widetilde{U} := \operatorname*{argmin}_{U \in \mathbb{R}^{n \times k}} \left\|(U\widehat{V} - A^T)\Psi\right\|_p,$$

$$\widehat{U} := \operatorname*{argmin}_{U \in \mathbb{R}^{n \times k}} \left\|(U\widehat{V} - A^T)\Psi\right\|_F,$$

$$U' := \operatorname*{argmin}_{U \in \mathbb{R}^{n \times k}} \left\|U\widehat{V} - A^T)\right\|_p$$

with solutions $\widetilde{U}, \widehat{U}$, and $U'$. Then using Lemma 8 and the fact that $\left\|U'\widehat{V} - A^T)\right\|_p \leq$ $\left\|U\widehat{V} - A^T)\right\|_p$ for all $U$ (and in particular, $U^*$), we have

$$\left\|\widetilde{U}\widehat{V} - A^T)\right\|_p \leq O(\log n) \left\|U'\widehat{V} - A^T)\right\|_p \qquad \text{(Lemma 8)}$$

$$\leq O(\log n) \left\|(U^*\widehat{V} - A^T)\right\|_p. \qquad \text{(minimality)} \qquad (16)$$

We know that $\widehat{U}_{i:} = A_{:i}^T \Psi(\widehat{V}\Psi)^{\dagger} = \min_x \left\|(x\widehat{V} - A_{i:}^T)\Psi\right\|_F$. Using Holder's inequality and equation (16), we have

$$\left\|(\widehat{U}\widehat{V} - A^T)\right\|_p = \sum_{i=1}^{n} \left\|(\widehat{U}_{i:}\widehat{V} - A_{i:}^T)\right\|_p$$

$$\leq \sqrt{\psi} \sum_{i=1}^{d} \left\|(\widetilde{U}_{i:}\widehat{V} - A_{i:}^T)\right\|_p \qquad \text{(Lemma 18)}$$

$$= \sqrt{\psi} \left\|\widetilde{U}\widehat{V} - A^T)\right\|_p$$

$$\leq O(\sqrt{\psi} \log n) \left\|(U^*\widehat{V} - A^T)\right\|_p, \qquad (17)$$

where the last inequality follows from equation (16). Substituting the value of $\widehat{U} = A^T \Psi(\widehat{V}\Psi)^{\dagger}$,

$$\left\|A^T \Psi(\widehat{V}\Psi)^{\dagger}\widehat{V} - A^T\right\|_p \leq O(\log d \log n \sqrt{\phi\psi}) \left\|U^*V^* - A^T)\right\|_p$$

$$+ O(\sqrt{\phi\psi} \log d \log n) \left\|\Phi^{\dagger} N_1 T^{\dagger}\right\|_p$$

Recall that $Y_c = A^T\Psi + N_2$ by construction in the algorithm. Using subadditivity of norms and substituting $\widehat{V} = (\Phi U^*)^\dagger \Phi A^T$, we have

$$
\begin{aligned}
\left\| Y_c(\widehat{V}\Psi)^\dagger \widehat{V} - A^T \right\|_p &= \left\| Y_c(\widehat{V}\Psi)^\dagger (\Phi U^*)^\dagger \Phi B - A^T \right\|_p \\
&\leq \left\| A^T\Psi(\widehat{V}\Psi)^\dagger \widehat{V} - A^T \right\|_p + \left\| N_2(\widehat{V}\Psi)^\dagger \widehat{V} \right\|_p \\
&\leq O(\sqrt{\phi\psi}\log d \log n)\left\| U^*V^* - A^T) \right\|_p + \left\| N_2(\widehat{V}\Psi)^\dagger \widehat{V} \right\|_p \\
&\quad + O(\sqrt{\phi\psi}\log d \log n)\left\| \Phi^\dagger N_1 T^\dagger \right\|_p \quad\quad\quad (18)
\end{aligned}
$$

Now,

$$
\left\| Y_c(\widehat{V}\Psi)^\dagger \widehat{V} - B \right\|_p \leq \left\| Y_c(\widehat{V}\Psi)^\dagger \widehat{V} - A^T \right\|_p + \left\| \Phi^\dagger N_1 T^\dagger \right\|_p
$$

Combining the above two inequalities, we get

$$
\begin{aligned}
\left\| Y_c(\widehat{V}\Psi)^\dagger \widehat{V} - B \right\|_p &\leq O(\sqrt{\phi\psi}\log d \log n)\left\| U^*V^* - A^T) \right\|_p + \left\| N_2(\widehat{V}\Psi)^\dagger \widehat{V} \right\|_p \\
&\quad + O(\sqrt{\phi\psi}\log d \log n)\left\| \Phi^\dagger N_1 T^\dagger \right\|_p
\end{aligned}
$$

Further, since $U^*$ has rank at most $k$, we have that $\widehat{V} = (\Phi U^*)^\dagger \Phi B$ has rank at most $k$. This implies that

$$
\begin{aligned}
\min_{\mathsf{rank}(X)\leq k} \| Y_c X B - B \|_p &\leq \left\| Y_c(\widehat{V}\Psi)^\dagger (\Phi U^*)^\dagger \Phi B - B \right\|_p \\
&\leq O(\sqrt{\phi\psi}\log d \log n)\left\| U^*V^* - A^T) \right\|_p + \left\| N_2(\widehat{V}\Psi)^\dagger \widehat{V} \right\|_p \\
&\quad + O(\sqrt{\phi\psi}\log d \log n)\left\| \Phi^\dagger N_1 T^\dagger \right\|_p \quad\quad\quad (19)
\end{aligned}
$$

**Third fundamental technique.** The last fundamental technique that we use is that an approximate solution of low-rank problem in the projected space also gives an approximate solution of the original low-rank problem. Let $R = PU\Sigma(\Phi PU\Sigma)^\dagger \Phi$, where $\widehat{X} = V_c\Sigma_c^\dagger[U_c^T Z V_r^T]_s \Sigma_r^\dagger U_r^T$. Let $\bar{X} = \mathrm{argmin}_X \left\| \Phi(PY_c\widehat{X}X\Phi B - B)T \right\|_1$. We have the following:

$$
\begin{aligned}
\| \Phi(RB - B)T \|_1 &\leq \sqrt{st}\left\| \Phi(PU\Sigma(\Phi PU\Sigma)^\dagger \Phi B - B)T \right\|_F &&\text{(definition of } P) \\
&= \sqrt{st}\min_X \| \Phi(PU\Sigma X\Phi B - B)T \|_F &&\text{(by definition of normal form)} \\
&\leq \sqrt{st}\left\| \Phi(PU\Sigma\bar{X}\Phi B - B)T \right\|_F \\
&\leq \sqrt{st}\left\| \Phi(PU\Sigma\bar{X}\Phi B - B)T \right\|_1 \\
&= \sqrt{st}\min_X \| \Phi(PU\Sigma X\Phi B - B)T \|_1 . &&\text{(definition of } \bar{X})
\end{aligned}
$$

This implies that $(\Phi U\Sigma P)^\dagger$ is the approximate solution of $\min_X \| \Phi(PU\Sigma X\Phi B - B)T \|_1$. Using Lemma 9, we have

$$
\left\| PU\Sigma(\Phi PY_c\widehat{X})^\dagger \Phi B - B \right\|_p \leq \sqrt{st}\min_X \| PU\Sigma X\Phi B - B \|_p
$$

Let $\bar{\Phi} = \begin{pmatrix} \Phi_k \\ 0 \end{pmatrix}$ and $\Phi'$ be the matrix such that $\Phi'_{i:} = \bar{\Phi}_{\pi(i):}$. Then we have the following set of inequalities

$$\min_X \|PU\Sigma X\Phi B - B\|_p \leq \left\|\frac{\phi}{k}\Phi^\dagger(U\Sigma)^\dagger(U\Sigma)\Phi Y_c(\widehat{V}\Psi)^\dagger\Phi B - B\right\|_1 \qquad \text{(by minimality)}$$

$$\leq O(\log d)\left\|\frac{\phi}{k}\Phi\Phi^\dagger(U\Sigma)^\dagger(U\Sigma)\Phi Y_c(\widehat{X}\Psi)^\dagger\Phi B - \Phi B\right\|_1 \quad \text{(no-dilation property)}$$

$$\leq O(\log d)\left\|\frac{\phi}{k}(U\Sigma)^\dagger(U\Sigma)\Phi Y_c(\widehat{X}\Psi)^\dagger\Phi B - \Phi B\right\|_1 \qquad (\Phi\Phi^\dagger = \mathbb{I})$$

$$= O(\log d)\left\|\frac{\phi}{k}\Phi' Y_c(\widehat{V}\Psi)^\dagger\Phi B - \Phi B\right\|_1 \qquad \text{(definition)}$$

$$\leq O(\log^2 d)\left\|Y_c(\widehat{X}\Psi)^\dagger\Phi B - B\right\|_1 \qquad \text{(Lemma 19)}.$$

Combining this with equation (19) and using the value of $\Pi$ gives

$$\left\|\Pi A^T - A^T\right\|_p \leq O(\log^3 d\log n\sqrt{st\phi\psi})\left\|U^*V^* - A^T)\right\|_p + O(\sqrt{st}\log^2 d)\left\|N_2(\widehat{V}\Psi)^\dagger\widehat{V}\right\|_p$$

$$+ O(\sqrt{st}\log^2 d)\left\|Y_c\widehat{X}(\Phi Y_c\widehat{X})^\dagger N_1 T^\dagger\right\|_p + O(\sqrt{st\phi\psi}\log^3 d\log n)\left\|\Phi^\dagger N_1 T^\dagger\right\|_p \tag{20}$$

All that remain is to bound each of the above additive term. The following claim does this.

**Claim 21.** *With probability at least* $97/100$,

$$\left\|N_2(\widehat{V}\Psi)^\dagger\widehat{V}\right\|_p \leq \widetilde{O}(kd\log n/\varepsilon),$$

$$\left\|\Phi^\dagger N_1 T^\dagger\right\|_p \leq \widetilde{O}(C_s C_t st/\varepsilon),$$

$$\left\|Y_c\widehat{X}(\Phi Y_c\widehat{X})^\dagger N_1 T^\dagger\right\|_p \leq \widetilde{O}(C_t st/\varepsilon).$$

*Proof.* Now $N_2(\widehat{V}\Psi)^\dagger\widehat{V}\Psi = \widehat{N}_2$. Using no dilation property of $\Psi$, we have $\left\|N_2(\widehat{V}\Psi)^\dagger\widehat{V}\right\|_p \leq \log n\left\|\widehat{N}_2\right\|_p$ This can be bound using the standard tail inequality for Laplace mechanism, i.e, with probability at least $99/100$, $\left\|\widehat{N}_2\right\|_p = \widetilde{O}(kd)$. Similarly, $\Phi Y_c\widehat{X}(\Phi Y_c\widehat{X})^\dagger N_1 T^\dagger T = \widehat{N}_1$ and $\Phi\Phi^\dagger N_1 T^\dagger T = N_1$. Using dilation and contraction properties of $\Phi, \Psi$, and $T$ completes the proof of claim. $\qquad\square$

We finish the proof by proving that the projection matrix is an orthonormal projection matrix with high probability.

**Claim 22.** $PU\Sigma(\Phi PU\Sigma)^\dagger\Phi$ *is an orthonormal projection matrix with probability* $99/100$.

*Proof.* Since $\Phi$ is a Cauchy matrix with i.i.d. entries, $\Phi$ is a full row matrix with probability $99/100$. Therefore, it follows from the definition of $P$ that

$$\Pi = PU\Sigma(\Phi PU\Sigma)^\dagger\Phi$$

$$= \Phi^\dagger(U\Sigma)^\dagger U\Sigma(\Phi\Phi^\dagger(U\Sigma)^\dagger U\Sigma)^\dagger\Phi$$

$$= \Phi^\dagger(U\Sigma)^\dagger U\Sigma((U\Sigma)^\dagger U\Sigma)^\dagger\Phi$$

$$= \Phi^\dagger(U\Sigma)^\dagger U\Sigma(\Phi^\dagger(U\Sigma)^\dagger U\Sigma)^\dagger$$

with probability $99/100$. This completes the proof. $\qquad\square$

The next proposition follows from the definition of $\widehat{X}$ and the fact that $N, N_1$, and $N_2$ are i.i.d. Laplace matrix.

**Proposition 23.** $Y_c\widehat{X}$ *has rank-$k$ with probability at least $1 - \delta$, where the probability is over the randomness of the algorithm.*

Using Claim 21 in equation (20) completes the proof of Theorem 15. $\qquad\qquad\qquad\square$

## C  Local Learning

Local differential privacy, a stronger variant of privacy, has gained a lot of attention recently. For e.g., it is the privacy guarantee employed by Apple in their new iOS [1] and has been used by Google for various data analysis [15]. In the local privacy model, there is no central database of private data. Instead, each individual has its own data element (a database of size one), and sends a report based on its own datum in a differentially private manner. The local model allows individuals to retain control of their data since privacy guarantees are enforced directly by their devices. However, it entails a different set of algorithmic techniques from the central model. In principle, one could also use cryptographic techniques to simulate central model algorithms in a local model, but such algorithms currently impose bandwidth and liveness constraints that make them impractical for large deployments.

Formally, we consider the database $X = [x_1, \cdots, x_n]^\top$ as a collection of $n$ elements (rows) from some domain $\mathcal{X} \subseteq \mathbb{R}^d$, with each row held by a different individual. The $i^{th}$ individual has access to $\varepsilon_i$-*local randomizer*, $R_i : \mathcal{X} \to W$ which is an $\varepsilon_i$-differentially private algorithm that takes as input a database of size $n = 1$. We assume that the algorithms may interact with the database only through local randomizers. We can then define local differential privacy as follows [13]. An algorithm is $\varepsilon$-locally differentially private if it accesses the database $X$ via the local randomizers, $R_1(x_1), \cdots, R_n(x_n)$, where $R_i$ is an $\varepsilon_i$-local randomizer, and $\max\{\varepsilon_1, \cdots, \varepsilon_n\} \leq \varepsilon$.

We note that what we have defined above is a non-interactive local differential privacy algorithm where an individual only sends a single message to the server. Another well studied variant is that of interactive local differential privacy where the server sends several query messages, each to a subset of users. Each such message, together with responses from users, counts as a *round* of interaction. In the end, the server aggregates and summarizes the messages it received from every user (over possibly multiple rounds), and uses it to answer queries about the data. It was argued in [27] that from an implementation point of view, it is more desirable to have as few rounds of interactions as possible because interaction introduces latency, synchronization, and bandwidth issues. In fact, existing large-scale deployments [1, 15] are limited to one that are noninteractive. Therefore, we limit our study to what is possible in the noninteractive variant of local differential privacy. We study robust principal component analysis in local model of differential privacy. We show that with high probability, we have that $\|A\Pi - A\|_p \leq \ell \cdot \mathsf{OPT}_k(A) + \widetilde{O}(\varepsilon^{-1}knd)$.

Our result is applicable in the setting when $\|A\|_p \gg O(nd)$. We note that, in practice, robust LRA is used on corrupted data matrix with a reasonable fraction of entries corrupted by large values. There are other scenarios, like network analysis, where private matrices have large entries. In such scenarios, typically $\|A\|_p \gg O(nd)$, and outputting an all zero matrix would incur an error far greater than what we incurred. If we wish to output a rank-$k$ matrix with provable guarantees, the naive algorithm that works as follows: every user add Laplace vector to their data and send the report to the server, and the server runs a non-private algorithm leads to worse additive error. This is because the low-rank approximation is now done on $A + N$ for $N \sim \mathsf{Lap}(0, 1/\varepsilon)^{n \times d}$. We next show that we can convert ROBUST-PCA to the model of local differential privacy. See Figure 3 for details. Our algorithm is non-interactive; therefore, we can use the generic transformation of [6] to get an $\varepsilon$-local differentially private algorithm.

**Theorem 24.** *Algorithm* LOCAL-ROBUST-PCA *(see Figure 3) is an $\varepsilon$-local differentially private algorithm. Furthermore, given a matrix $A \in \mathbb{R}^{n \times d}$ with $\mathsf{OPT}_k(A) := \min_{\mathsf{rank}(X) \leq k} \|A - X\|_p$,* LOCAL-ROBUST-PCA *runs in time* $\mathsf{poly}(k, n, d)$*, space* $\widetilde{O}(k(n+d))$*, and outputs a rank $k$ projection matrix $\Pi$ such that, with probability $9/10$ over the randomoziation of the algorithm,*

$$\|A - A\Pi\|_p \leq O(\log n \log^3 d \, (k \log k \log(1/\delta))^{2(2-p)/p})\mathsf{OPT}_k(A) + \widetilde{O}(k^2nd/\varepsilon).$$

---

**Algorithm 3** LOCAL-ROBUST-PCA

---

**Input:** Every user $i \in [n]$ having access to a row $A_i$, target rank $k$

**Output:** Rank-$k$ projection matrix $\Pi \in \mathbb{R}^{d \times d}$

1: **Initialization.** Sample $\Phi \in \mathbb{R}^{\phi \times d}, \Psi \in \mathbb{R}^{n \times \psi}, S \in \mathbb{R}^{s \times d}$, and $T \in \mathbb{R}^{n \times t}$ with every entry sampled iid from $\mathcal{D}_p$. All these matrices are publicly available.

2: **user side computation:** every user $i$, **do**

3:    **Sample private noise.** Sample $N_{1,i} \sim \mathsf{Lap}(0, C_\phi C_t)^{\phi \times t}, N_{0,i} \sim \mathsf{Lap}(0, C_\psi)^{d \times \psi}, N_{2,i} \sim \mathsf{Lap}(0, C_s C_t)^{s \times t}$ using its private coin.

4:    **Construct.** $A^{(i)} \in \mathbb{R}^{d \times n}$ with all zero entries except the column $i$ has entry $A_i$.

5:    **Compute.** $Y_{r,i} = \Phi A^{(i)} T + N_{1,i}, Y_{c,i} = A^{(i)} \Psi + N_{0,i}$ and $Z_i = Y_{r,i}$, where $\Phi, T$ and $\Psi$ are sketching matrices with every entries sampled i.i.d. from a $p$-stable distribution.

6:    **Send.** $(Y_{r,i}, Y_{c,i}, Z_i)$ to the server.

7: **end user side computation:**

8: **server's computation:** get $\{Y_{r,i}, Y_{c,i}, Z_i\}_{i=1}^n$, **do**

9:    **Compute.** $Y_c = \sum Y_{c,i}, Y_r = \sum Y_{r,i}$, and $Z = \sum Z_i$.

10:    **Compute.** $\mathsf{SVD}(\Phi Y_c) = [U_c, \Sigma_c, V_c]$. and $\mathsf{SVD}(Y_r) = [U_r, \Sigma_r, V_r]$.

11:    **Set.** $\widehat{X} = V_c \Sigma_c^\dagger [U_c^T Z V_r^T]_k \Sigma_r^\dagger U_r^T$, where $[B]_k = \operatorname{argmin}_{r(X) \leq k} \|B - X\|_F$.

12:    **Pick:** a permutation matrix $Q \in R^{\phi \times \phi}$.

13:    **Compute:** the full SVD of $Y_c \widehat{X}, [U', \Sigma', V']$. Set $U = U'Q, \Sigma = \Sigma'Q$, and $P = \Phi^\dagger (U\Sigma)^\dagger$.

14:    **Output:** $\Pi = PU\Sigma(\Phi PU\Sigma)^\dagger \Phi$.

15: **end server's computation:**

---

Note that the naive algorithm that works as follows: every user add Laplace vector to their data and send the report to the server, and the server runs a non-private algorithm leads to a slight worse additive error. This is because the low-rank approximation is now done on $A + N$ for $N \sim \mathsf{Lap}(0, 1/\varepsilon)^{n \times d}$. Song et al. [29] would then imply that the additive error would be about $O(C_\phi C_\psi \sqrt{st\psi\phi}nd)$.

*Proof of Theorem 24.* Using the same arithmetic as in the proof of Theorem 15, the error incurred would be

$$\|A - A\Pi\|_p \leq O(C_\phi C_\psi \sqrt{st\phi\psi})\mathsf{OPT}_{k,p}(A) + c_2\left(2n\left\|\widetilde{N}_1\right\|_p + 2n\left\|\widetilde{N}_2\right\|_p\right),$$

where $\Pi := PU\Sigma(\Phi PU\Sigma)^\dagger \Phi$, $\mathsf{OPT}_{k,p}(A) := \|U^*V^* - A)\|_p$, $N_1$ is an $n$ times a $d \times \psi$ random Laplace matrix, $\widetilde{N}_1$ and $\widetilde{N}_2$ are as formed in the proof of Theorem 15. Using the same calculation completes the proof of Theorem 24.   $\square$

## D   A Closer Look on Current Techniques

We first give the argument we made in the main text for the accuracy guarantees by using the $\zeta$-net mechanism of Blum et al. [4]. To apply Blum et al. [4] in our setting, we need to compute the number of k-tuples of unit vectors in $\mathbb{R}^d$ and $\mathbb{R}^n$. The size of $\zeta$-net of unit vectors in $\mathbb{R}^d$ (for row space) is $p = \zeta^{1-d}$. Hence, number of $k$-tuples of unit vectors is $\binom{p}{k}$. Similarly, for column space, it is $\binom{\zeta^{1-n}}{k}$. This gives us the error claimed earlier in the introduction.

There are two main approaches for efficient private algorithms – output perturbation and input perturbation. In output perturbation, we first compute the output (e.g. rank-$k$ approximation of a given matrix) non-privately and then add appropriately scaled noise to preserve privacy. In input perturbation, we add noise to the private matrix and then compute the output on the noisy matrix. Both these approaches require adding noise to every entry of the given input matrix or to every entry of the non-private output matrix. Consequently, both of these methods would incur an additive error of $O(nd)$.

Alternatively, one could consider iterative approaches, such as noisy Krylov subspace iteration [19], for finding low-rank matrix approximation with respect to spectral norm. However, it is not immediately clear how to adapt such an algorithm for $\ell_p$ low-rank approximation. The methods used in known results for differentially private low-rank approximation with respect to entrywise $\ell_2$-norm,

say [18, 14, 32], also has some hurdles. The main problem here is that the given objective is not rotationally invariant. If we just use the output produced in any of the results for Frobenius norm and then use, say Holder's inequality, then the accuracy would depreciate proportional to $(nd)^{1/p-1/2}$.

One may then argue that we can solve robust low-rank approximation for constant dimension by using *exponential mechanism* [26]. For using exponential mechanism, we need to find a suitable *scoring function*, which is not clear in the case of entrywise $\ell_p$-norm. Even if we are able to find a scoring function analogous to one used in [21], it is not clear whether we can iterate it for $k$ rounds to get all the top-$k$ subspace. More precisely, it is not clear whether a result analogous to the Deflation lemma of [21] holds in the case of entrywise $\ell_p$ approximation.