[Reviews · NeurIPS 2018]

Reviewer 1



This paper studies the problem of low-rank approximation in the l_p norm, while guaranteeing differntial privacy. The verion of the probl em when we care about the usual frobenious norm (or any rotationally invariant norm) approximation is the usual PCA and DP PCA has been t he subject of intense study, with tight upper and lower bounds known for various reasonable privacy models. In the non-private setting, t here has been some recent work on provable bounds on the best low-rank approximation in terms of entry-wise l_p norm of the residual. The current submissio takes those algorithms, and find appropriate differentially private variations of those. This is a mathematically challenging task, as it is not usually clear what the best way is of making an algorithm differentially private. The current paper shows that under their definition of privacy, an appropriate choice is to use random sketchs as in the non-DP algorith ms for this problem, and once the problem has been reduced down to that of finding a low-rank approximation of a polylogarithmic sized ma trix, one can use the usual PCA since for small matrices, all norms are related by factors polynomial in the dimension. The main tool is to use a differentially private PCA at this step, and to make this all work, the authors show how to analyze the sensitivity of the matri x one is doing this PCA when moving between neighboring datasets in the privacy definition. I find the utility guarantee of the main algorithm somewhat weak wrt the privacy definition. Specifically, the privacy definition only protects changes to the matrix by an l_1 norm of 1. A priori, this is a reasonable privacy definition. E.g. if the matrix of interest is a covariance matrix over features and each user has a sparse small-norm feature vector then one can imagine that each user has limited influence in terms of the l_1 norm. However, the approach incurs an additive overhead of $O(k(n+d))$ in this setting, which seems rather larg e. It would be useful if the authors compared this to what is obtainable, and e.g., what one pays in the usual PCA setting. Further, the authors would do well to explain better where this privacy neighborhood relation makes sense. It seems to me that in most se ttings where I would be ok with edge-level privacy, the true goal is to get a handle on the right (or left) singular values, rather than getting a low-rank representation of the matrix itself that is private. I have now read the rebuttal and am convinced that the privacy definition is a reasonable one. I would encourage the authors to improve the writeup and justify the privacy definition better.

Reviewer 2



SUMMARY: This is a theory paper proposing a new algorithm for low-rank matrix approximation (LRA) that satisfies feature-level, approximate (delta > 0) differential privacy (DP). (The 'robust' in the title simply refers to using the pointwise l_p norm with 1<=p<2 as the LRA loss function, which is a setting that does not appear to have been studied under the DP constraint before.) The paper analyses the multiplicative and additive error of the proposed algorithm. Content-wise the paper attempts to provide a good description of (1) how the proposed algorithm fits into the landscape of existing (DP) LRA algorithms, (2) why existing algorithms fail in the 'robust' and DP setting considered here, and (3) what are the intuitions and key ideas behind the proposed algorithm. Unfortunately, the clarity of presentation is at times hindered by poor writing. (As I'm quite new to this field (especially matrix factorisation), please excuse any erroneous comments and please feel free to directly point them out.) MAIN COMMENTS: 1/ [significance] This could appear as another paper that takes an existing task X and simply looks at what happens when one wants to solve it under DP, proposing a new DP-X algorithm. However, I think X=LRA can be a particularly interesting problem to look at under the DP constraint, for the following reason. In DP we are constantly looking for ways to reduce the amount of noise required in order to reach a given privacy guarantee, and one folk wisdom general advice seems to be to avoid privatising high-dimensional objects (especially when some dimensions can carry little useful statistical information, but still need to be privacy-protected as we don't know a priori which dimensions these are). In studying DP-LRA one is forced to look at this question whether interest in a lower-dimensional description of the data allows one to get away with less noise being added, and this paper answers this question in the positive for the setting studied. I wonder if there might be some substance in this view and if it could help differentiate this paper from other DP-X papers. 2/ [significance] Is the proposed algorithm practical, or is it theoretical at this stage? Specifically, could you compute exact values of epsilon and delta when the algorithm is run? I can see there is an empirical evaluation in Appendix C, but I cannot find the corresponding values of epsilon and delta. 3/ [quality] The 'robustness' requirement appears to be only used as motivation for using the pointwise l_p norm with 1<=p<2 as the LRA loss function (which is known to impart some notion of robustness). However, the paper could be more explicit about the exact ranges of p in different statements. - Are all statements supposed to hold for 1<=p<2 only? (Some sentences, including one in the Abstract, specifically refer to the p=2 case as well.) - Which statements would fail for other values of p? 4/ [quality] Section 3.2 describing the extension to the streaming setting and in particular the section on local DP appears a bit too informal to me. For example, it would be helpful to state explicitly the privacy setup in the streaming setting (when are two streams of updates considered neighbouring?) and how the privacy loss would accumulate under the continuous release model. Would one simply use an (advanced) composition theorem from the DP literature, or can one do better here? 5/ [originality] To my knowledge the paper is novel, and it is well discussed against existing work, describing the differences in assumptions, theoretical results and algorithmic ideas used. 6/ [clarity] The clarity of writing could be substantially improved at places. The submission would also really benefit from just careful proof-reading for grammar (although I'm not penalising in my review for bad grammar, fixing even singular vs plural in many places would have made reading the submission easier). Please see also minor comments below for some concrete suggestions. 7/ [quality] Re line 174: Why would any privacy preserving mechanism need to necessarily privatise the matrix A? Perhaps it could instead select x using something like the Exponential mechanism, or privacy-protect C_i instead of A, or use a different approach entirely. STRENGTHS: [+] LRA seems to be an interesting problem to study under DP, and it appears the case considered here (pointwise l_p norm loss function with 1<=p<2) had not been studied yet. [+] Content-wise the paper is on the right track in providing the reader with necessary background, explaining differences to prior work, and the key ideas in constructing the new algorithm. WEAKNESSES: [-] Unclear whether the paper is theoretical, or if the proposed algorithm gives a meaningful privacy guarantee with reasonable accuracy. [-] The quality of the writing (and grammar) should be improved. [-] Being a theory paper first and foremost, a more suitable venue than NIPS could be a journal where the reviewing process scope would be such that the proofs (currenty in the Appendix) are also carefully peer-reviewed. MINOR COMMENTS: (12, 76) In robust subspace learning, wouldn't it make sense to restrict the X in the definition to projection matrices as well, so that an algorithm would have a chance of having no approximation error at least when k=d? (15) Is "measure" meant in a technical sense here? Also, the problem could potentially be generalised by considering general distances d(A, B) rather than just those stemming from norms via d(A, B) = ||A-B||. (21) What is meant by a "robust" privacy guarantee? Confusing given there is another notion of robustness at play here. (23) Missing object after "de facto". (24) If the output of M is not discrete, a rigorous definition would only consider measurable sets S, rather than all subsets of the range. (41) May want to define "efficient". (46) Based on line 224, Big-O with tilde also hides poly log factors in other variables (e.g. poly log k), not just in n. (47) Definition 2 needs to specify the range of p, otherwise there is no connection to robustness. (Equation 1) Tau in this equation was called gamma in the Abstract. (52) The value of alpha here doesn't depend on the privacy parameters epsilon, delta. Is this the best alpha one can achieve even in the non-private setting, or is there a hidden privacy cost here? (59) "l_p-norm" -> "l_1-norm" (60-62) Holder's inequality only seems to provide an *upper bound* on the error when measured in terms of the l_p norm for p<2 (using a different approach, one might obtain a tighter bound). So strictly speaking the "exponential improvement" would only apply to the p=2 case, which is however outside of the robustness scope considered? (66) I can see how feature selection relates to projection matrices, but I'm not sure how projection matrices are related to representation learning, where the learned representations need not correspond to projections. Can you please clarify? (70) "Problem 1" -> "Definition 2" (91) It is said that the result of Song et al. [37] holds even when an additive error o(k(n+d)) is allowed. However, this does not seem to subsume the case considered in this submission, where an O(k(n+d)) additive error is allowed. (106) "Table 1.2" -> "Table 1" (115) The sentence is false as written. (126) Missing brackets around v_1, ..., v_n. (133, 135) Just checking, intuitively I would have swapped the definitions of Dilation and Contraction. (Lemma 7) Variable naming conflict for "p", which is used to denote two different quantities. (150) I think it's a bit unfair to say the two main approaches to DP are input and output perturbation, as many DP algorithms operate by privacy-protecting an intermediate object in the computation pipeline (e.g. DP-SGD protects gradients). (184) Do you mean "c" instead of "s"? (246-255) Could also refer to Table 1 again, to compare the result obtained in the paper being described. (261) Should say A_{i_t, j_t} instead of A_{i_{t-1}, j_{t-1}}. ===== Update after author response and reviewer discussion ===== Thank you for the responses to our reviews. I'm happy with the answers to our technical comments, and I would like to suggest incorporating some of these clarifications into the next revision. Adding a comment on motivating the neighbouring relation (as requested by fellow Reviewer #1) would be particularly helpful. I don't think the quality of writing was good enough for NIPS in the submitted version of the paper, but I believe this can be fixed in the next revision. Please do have your paper proof-read for style and grammar issues. Assuming these changes are made, I've increased my score from 5 to 6.

Reviewer 3



The authors propose a differentially private algorithm for robust low-rank approximation that improves both the multiplicative approximation and the additive error significantly. They also adapt this algorithm to differentially private robust subspace learning where the additive error does not scale with the size of the dataset but the rank. It is a well-written, clear paper. The topic is important and the results are significant.